# Temporal Stages of Burnout: How to Design Prevention?

**DOI:** 10.3390/ijerph21121617

**Published:** 2024-12-03

**Authors:** Céline Leclercq, Isabelle Hansez

**Affiliations:** Human Resources Development Unit, Faculty of Psychology, Speech Therapy and Education Sciences, University of Liège, 4000 Liège, Belgium; ihansez@uliege.be

**Keywords:** burnout intervention, stages, prevention, return-to-work

## Abstract

Burnout, a major concern defined most commonly in the literature with a symptoms-based classification, can also be described as a temporal process with various stages experienced by workers, each composed of unique characteristics and challenges. This intricate process of burnout emphasizes pivotal moments, such as engagement and enthusiasm with a high job ideal (Stage 0), weakening of the ideal (Stage 1), protective withdrawal (Stage 2) and confirmed burnout (Stage 3). Through an opinion review, the objective of this article is to examine which prevention level, and more specifically which prevention actions, can be developed at each stage of this temporal process of burnout. The review criteria allow for the integration of both individual- and organization-focused interventions, ranging from early organizational-level strategies (primary prevention) to clinical consultations addressing the erosion of professional ideals (secondary prevention), as well as psychoeducational sessions aimed at promoting worker well-being. In conclusion, the article underscores the need for a holistic approach, combining organization-focused interventions with individual-focused interventions. Through a comprehensive review, this research provides insights into evidence-based practices, identifies gaps in current research, and offers guidance for future interventions for better support of workers facing burnout.

## 1. Introduction

In an era marked by rapid technological advancements and global market fluctuations, organizations face increasingly turbulent and competitive environments. Conceptual analyses have argued that such conditions demand flexible strategies to sustain business excellence, particularly by adapting to rapidly changing market forces [1]. Similarly, there is a need for strategic flexibility and innovation to navigate the intensified global competition driven by technological change [2]. Building on this context, predictions have been made that 47% of jobs in the USA are at high risk of automation within the next two decades, underscoring the potential for significant workforce disruption [3]. Although these figures have been contested by studies suggesting more conservative estimates [4], there is broad consensus that emerging technologies will substantially reshape the workforce, particularly affecting lower-skilled workers [5].

At the individual level, mixed-methods studies have revealed that digital platforms heighten job insecurity and stress among gig workers [6]. Similarly, research has shown that automation blurs work boundaries, contributing to increased stress and social isolation [7]. On a group level, it has been found that rapid technological change can erode team cohesion, leading to resistance to change [8]. Organizationally, it has been demonstrated that poorly managed digital transformations result in higher turnover and reduced performance [9].

These findings highlight the need for organizations to cultivate a resilient, high-performing workforce by creating supportive work environments [10]. Recent research has shown that job resources correlate positively with employee well-being, while other studies have emphasized that individual differences, such as resilience, significantly influence how workers adapt to job demands [11,12,13,14]. Thus, tailored approaches are essential for fostering well-being and productivity.

Work, while fulfilling basic psychological needs, can become a source of overwhelming stress, leading to burnout. Burnout is increasingly recognized as a critical issue impacting mental health and productivity, as evidenced by rising long-term sick leave due to psychological problems, including burnout, in Belgium. Statistics from the Belgian Government’s Federal Public Service Employment, Labour and Social Dialogue highlight that approximately 500,000 individuals are on long-term sick leave, with psychological issues accounting for 32.7% of these cases in 2022 according to Securex. In terms of public health costs, long-term sick leave is estimated at approximately 478 million euros in 2020, i.e., an increase of 39.92% since 2016 [15].

Similarly, at the European level, burnout prevalence varies considerably, influenced by factors such as work culture, employment conditions, and national occupational health policies [16]. Approximately 10% of the European workforce suffers from burnout, with significant variability in professions like medicine, where the prevalence can range from 7.7% to 43.2% [16]. Globally, burnout is also prevalent, particularly in developing regions [17]. Burnout among healthcare providers in sub-Saharan Africa ranges from 26% to 78%, driven by challenging work environments [17]. High burnout rates have also been reported among HIV service providers in South Africa and Zambia, largely due to intense workloads and limited support [18]. Similarly, about 31.5% of nurses in the United States experience significant burnout, especially in high-stress settings, like intensive care units [19]. These findings underscore the need for tailored occupational health interventions across different regions and professions.

These figures reveal a critical trend: not only is long-term absence becoming more common, but the role of burnout as a contributing factor is growing more pronounced, signaling an urgent need for effective preventive strategies and support systems to address this increasing problem.

### 1.1. Definitions and Temporal Stages of Burnout

The importance of burnout transcends its prevalence, affecting both individual and organizational levels. At the individual level, burnout can lead to long-term health issues, including cardiovascular and mental health problems [20]. At the organizational level, burnout is associated with reduced productivity, mistakes, low job satisfaction and engagement, as well as increased absenteeism and turnover [21].

The concept of burnout has evolved since its definition in the 1970s, when Freudenberger first described it as a gradual emotional exhaustion and loss of motivation among volunteers in aid organizations [22]. Nevertheless, this initial belief that burnout was confined to human-services sectors has been then dismissed, leading to a broader conceptualization applicable across various occupational fields [23,24,25]. Subsequent research by Maslach and colleagues expanded the understanding of burnout to involve exhaustion, depersonalization, and reduced personal accomplishment [26]. Despite extensive research, the academic community has not yet agreed on a singular definition of burnout. Canu and colleagues identify no fewer than 13 different definitions published over four decades [27]. Common to most definitions, however, is the central role of exhaustion, whether emotional, physical, cognitive, or a combination. The association of burnout with cognitive impairments has also been consistently observed, suggesting a broader impact on mental functioning [28]. Recently, burnout was recognized by the World Health Organization (WHO) and the 11th revision of the International Classification of Diseases (ICD-11) as an occupational phenomenon; however, they always distinguish this syndrome from medical conditions [29].

Recent researchers consider burnout as a dynamic process specifically focused on symptomatology [30,31]. They have found that burnout usually includes four core symptoms: exhaustion, emotional impairment, cognitive impairment, and mental distance. Stress is typically the first warning sign and the main reason people ask for help. Besides these, there are three more signs to watch out for: depressed mood, psychosomatic complaints, and psychological distress symptoms. Burnout often starts with a loss of the energy required to regulate emotional and cognitive processes. To protect themselves, workers mentally distance themselves and adopt a detached and cynical attitude towards work, which is perceived as the main source of burnout. However, this mental distancing is ineffective because it provokes negative reactions from others, compromising motivation at work and performance. This contributes to exacerbating burnout.

Over recent decades, the majority of self-reported burnout scales have also been created on the basis of these dimensions, in order to assess the risk of burnout in workers. Among others, these include the Maslach Burnout Inventory (MBI) based on the three dimensions, the OLdenbrug Burnout Inventory (OLBI) based on exhaustion and disengagement, and the Burnout Assessment Tool (BAT), recently developed by Schaufeli et al. based on their new definition, as seen above [31,32,33].

Another model, created through a narrative review, for understanding burnout focuses on both the progression of symptoms and the worker’s changing perception of his/her deteriorating relationship with his/her job [34,35,36]. This model is based on nine qualitative and descriptive studies and aims at identifying the burnout process through a multifocal (with a focus on both symptoms and the evolution of organizational and individual contexts) and a temporal perspective [37,38,39,40,41,42,43,44,45]. Burnout unfolds in a complex, multi-stage process that begins with what is known as ‘engagement and enthusiasm with a high job ideal’, or Stage 0. Initially, it is characterised by work engagement with idealistic enthusiasm. This is seen as the main source of personal accomplishment, so the worker invests much energy into his/her job, sometimes to the detriment of private life. The worker is motivated to achieve ambitious goals with positive energy and determination. At this level, work is idealised, and the signs of burnout are almost absent.

Stage 1, entitled ‘weakening of the ideal’, illustrates the reality of the job, with its contradictions and unexpected changes, which have an impact on the initial enthusiasm (Stage 1). Despite their investment until exhaustion, workers come up against obstacles that hinder their progress and their desire for fulfilment, leading to a feeling of stagnation and doubt.

Stage 2 marks a turning point with the emergence of ‘protective withdrawal’. Work, once a source of satisfaction, becomes a threat, and the worker attempts protective strategies to avoid harmful situations. This protective phase is accompanied by growing cynicism and a rejection of the organizational values once respected. The meaning of work is questioned, and work-related problems gradually have an impact on private life, with emerging physical, cognitive, and behavioural symptoms.

Finally, the process culminates in Stage 3, entitled ‘confirmed burnout’ [34,35,36]. Usually, before this stage, the worker is unaware of the problem, which is why he or she feels as though he or she has suddenly experienced burnout. However, proven burnout is the result of a long-term process associated with a form of denial. The critical incident and sick leave generally mark the end of denial. The ideal of a fulfilling job has disappeared, and the worker finds himself/herself unable to continue. Often, a critical incident signals the end of the resistance, accompanied by intense emotional and physical distress. The worker may find himself/herself off work, faced with ambivalent feelings and an increased risk of depression. Doubts extend around the worker’s entire identity. This sick leave is sometimes the beginning of an awareness and a necessary re-evaluation of one’s priorities and one’s relationship with work.

Through this progression, burnout reveals itself not just as a state but rather as a process, where the changing relationship between the worker and his/her work environment can lead to a gradual erosion of work engagement and well-being. Understanding this process is crucial for effective diagnosis and intervention, highlighting the need for attention to both the personal experiences of workers and the broader organizational context [34,35,36].

### 1.2. Burnout Prevention

Preventive measures against burnout can be distinguished between individual-focused and organization-focused interventions [46].

Individual-focused interventions aim to empower workers by developing personal coping mechanisms, such as resilience training, stress management programs, and mindfulness practices [47,48,49]. These interventions are designed to enhance an individual’s capacity to manage stress and to maintain a healthy work–life balance. For instance, some authors found that individual interventions can have positive, though generally moderate, effects on burnout symptoms and employee well-being [50,51]. However, other authors, such as Maslach et al. [52] and, more recently, West et al. [53], argue that individual strategies for coping with workplace stress are frequently ineffective on their own. They suggest that employees frequently lack control over the sources of stress within their work environment. Consequently, while individual techniques for managing stress may offer short-term relief, they might not address the underlying systemic issues causing burnout. Generally, individual interventions show effectiveness in the short term, up to 6 months, but maintaining positive effects requires ongoing support [47].

On the other hand, organization-focused interventions involve creating a supportive work environment that actively mitigates the risk factors associated with burnout [52]. This includes implementing work procedures, like task restructuring, work evaluation and supervision [54]. In addition, leadership training is essential to equip direct supervisors with the skills they need to recognise the early signs of burnout and to support their teams effectively [55]. These interventions tend to have more sustainable effects, with some studies showing benefits lasting up to a year [54]. By addressing the structural causes of stress, organizational interventions can lead to enduring improvements in employee well-being [50]. However, it is also noted that the effectiveness of organizational interventions alone might diminish over time if not complemented by individual-focused approaches [54].

The limitation of single-level approaches—whether individual or organizational—stems from their inability to address both individual and systemic factors comprehensively [47]. The literature suggests that combining both individual and organization-focused interventions is the most effective strategy for preventing burnout [47,54,56]. This combined approach is particularly advantageous because it addresses both personal coping mechanisms and systemic issues contributing to burnout. Such a dual strategy significantly reduces burnout symptoms and facilitates better return-to-work outcomes compared to single-level strategies [47,56]. Combining individual and organizational interventions not only enhances short-term relief but also provides a more robust and sustainable solution by integrating personal resilience with systemic changes [56].

Moreover, early intervention is key. Organizations must be vigilant in identifying the signs of burnout, particularly in the initial stages, where workers may still be highly engaged but beginning to experience the mismatch between their job ideals and the realities of their work environment (Stage 0 to 1) [34,35,36].

Traditionally, prevention is based on three levels: primary, secondary, and tertiary [57,58].

First, the primary prevention mainly focuses on collective organization-based interventions to identify and to eliminate or reduce stressors before they lead to burnout (cf. the Job Demands–Resources model) [59]. This involves creating a healthy and engaging work environment with policies such as promoting work–life balance, recognizing worker achievements, and encouraging open communication.

The next level is secondary prevention and is concerned with the early detection of signs of stress and exhaustion and rapid intervention to prevent the evolution of burnout and to maintain workers in their jobs [46]. This might include regular assessments of worker well-being, stress management programs, and resilience training. For example, a Belgian pilot project, conducted by FEDRIS (Federal Agency for Occupational Risks), tested a Burnout Treatment Program in hospital and banking sectors for workers at an early stage of burnout [60].

Finally, the tertiary prevention is mainly based on individual-focused interventions, when workers already suffer from burnout [47]. This stage of prevention aims to minimize long-term damage while facilitating recovery and reintegration into the workplace. Interventions may include psychological support, professional reintegration programs, and adjustments to professional responsibilities. At this stage, the focus is on healing and rehabilitation, with personalized support to help workers regain their health and effectiveness at work [57,58].

By integrating these three levels of prevention, organizations can develop a robust framework for not only preventing burnout but also for managing it effectively when it occurs. This requires close collaboration between workers, direct supervisors, and prevention advisors, as well as a deep understanding of how individual and organizational factors interact to influence well-being at work. The prevention of burnout is an ongoing process that requires a concerted effort from both individuals and organizations [47,54,56]. By understanding the temporal stages of burnout and implementing a dual approach (based on an individual and an organizational approach) to prevention, organizations can not only safeguard the well-being of their workers but also enhance their overall performance and productivity [59].

The objective of this article is to explore, through an opinion review, the different levels and types of prevention actions that can be implemented at various stages of the burnout process. This opinion review emphasizes the importance of an inclusive approach that combines individual- and organization-focused strategies. The goal is to propose a flexible prevention framework that can be tailored to the specific needs and constraints of each context, thereby supporting a comprehensive and adaptable management of burnout.

## 2. Methodology

This paper reviews research on preventing burnout by examining the evolution of this issue. An opinion review format was chosen for its flexibility, which allows for integration of the viewpoint of a scientific and field expert and for a wide-ranging inclusion of articles and the incorporation of professional insights and viewpoints. This review is based on more than 30 years in scientific research in the field of occupational psychology and more than 150 psychosocial aspects-related interventions in organizations. The research was primarily sourced from scientific databases and Google Scholar, with further studies identified through the references of relevant articles. This allowed us to identify a wide range of research, by integrating regularly updated and accessible articles and pratical documents from the field. The search was not limited by publication date, but nearly all selected studies were published after the year 2000. Search terms as ‘burnout prevention’, and ‘burnout treatment’ were used in different combinations to ensure a comprehensive search.

The selection of studies for review was a step-by-step process. It started with reviewing titles and summaries to identify those relevant to burnout prevention. Next, the full text of these articles was examined to determine if they met specific criteria: they had to be research studies, reviews, or theoretical papers focused on measures, strategies, or interventions for preventing burnout. Exclusion criteria concern documents that are not fully accessible, either through open access or via institutional subscriptions, as well as non-relevant contexts (e.g., parental or school contexts), in order to maintain a focus solely on the issue of burnout from a professional standpoint as recognized by the WHO. A total of 65 studies [37,47,48,49,50,54,59,60,61,62,63,64,65,66,67,68,69,70,71,72,73,74,75,76,77,78,79,80,81,82,83,84,85,86,87,88,89,90,91,92,93,94,95,96,97,98,99,100,101,102,103,104,105,106,107,108,109,110,111,112,113,114,115,116,117] were used to discuss and illustrate the key moments in burnout prevention, based on long experience in the field and integrating both an evidence-based approach and the requirements of national legislation on well-being at work (that of, France and Belgium in particular) compelling employers to implement preventative actions.

The method allowed for the combination of findings from various types of research, which shows how complex and layered burnout prevention is. It also includes practical knowledge from our field experience, offering insight into the real-world effectiveness of the prevention discussed.

How can we use the stages of burnout to think about prevention? Defining burnout as a process makes it possible to consider avenues of prevention according to the stages described. Based on the characteristics of the different stages, we believe that it is relevant to define, for each stage, the type of prevention/intervention to be prioritized. We have based our thinking on four key moments requiring a specific type of prevention: (1) upstream of professional career, when the professional ideal is being built up (Stage 0); (2) during idealized work engagement (Stage 0), then during weakening of the ideal (Stage 1); (3) at the onset of protective withdrawal (Stage 2); and, finally, (4) when burnout is confirmed (Stage 3). Based on the definition of types of prevention [57,58], the first two key moments, i.e., the construction of the ideal and the idealized work engagement, fall under the logic of primary prevention. Prevention aimed at workers who had begun to doubt their ideal, or to set up protective mechanisms, would fall within the logic of job retention or secondary prevention. Finally, for workers with a confirmed burnout diagnosis, the final stage in the process should be supported by tertiary prevention, including a return-to-work process. In the following paragraphs, we describe in greater detail the preventive measures envisaged for the four key moments. It should be noted, however, that an intervention proposed at one key moment may also be relevant at a later one. In the same way, secondary or tertiary prevention does not exclude the need to again activate primary prevention, which is essential for integrating the collective dimension.

## 3. Key Moments for Burnout Prevention According to the Stages of Burnout

### 3.1. Key Moment 1: Constructing the Ideal (Stage 0)

An initial approach involves intervening in the primary education of future professionals, focusing on sensitizing them to the realities of specific professions and their associated risks, especially psychosocial risks. It seems relevant during training courses to raise awareness about the potential gap between the profession’s ideal and the on-ground reality. Through a longitudinal qualitative study, such programs have already shown that sensitizing nursing students during their training to these challenges is effective. While many professionals enter the field with strong, idealistic values, organizational and professional constraints often compromise these ideals, leading to frustration and burnout. However, those who work in supportive environments are better able to maintain and implement their ideals, underscoring the need for educational programs that both prepare students for these realities and promote environments where their ideals can flourish [61].

Primary interventions concerning the ideal can also be initiated by organizations immediately from the recruitment and selection phase of workers. It is quite easily applicable that recruiters offer candidates a balanced view of both the positive and negative aspects of vacant jobs, as well as detailed information about the culture and policies of the recruiting organization. This ensures that candidates are informed in advance about the conditions under which they will work. This approach aligns with a technique known in personnel selection as a ‘Realistic Job Preview’. This relies on various methods, such as video presentations, work simulations, meetings with current workers, and informational brochures [62]. This strategy allows the worker to prepare for his/her new role and to adjust his/her expectations more in line with what the job can offer. It has been shown to have a moderate yet effective impact, reducing turnover rates (19% of those exposed to RJPs had quit within the first six months, compared to 27% in the control group), improving job satisfaction, and lowering intentions to quit [62]. Thus, this approach could minimize the gap between the professional ideal and reality.

Lastly, another process related to the ‘Needs–Supply Fit’ (NS) concept [63] used during personnel selection is the analysis of the match between a candidate’s expectations and what the organization can provide to meet those expectations. Ensuring a good fit during the selection stage, or an internal mobility process, using, for instance, a tool that provides a multi-dimensional view on the analysis of twelve worker needs, such as stimulating work, a good work–life balance, predictable work schedules, and flexibility practices [64], have proven to have several benefits for the organization. For example, research has shown that a strong NS fit is positively correlated with increased organizational identification, which in turn enhances job performance and promotes organizational citizenship behaviors. These outcomes underscore the value of achieving a good person–environment fit to facilitate integration [65] and retention of newcomers in the organization [63], increase job satisfaction [66] and performance [67], and reduce health-related risks [68].

### 3.2. Key Moment 2: Engagement and Enthusiasm with a High Job Ideal (Stage 0)

To promote and maintain well-being at work is essential to develop a dynamic risk management policy within the framework of national policies (for instance, in Belgium, the 2014 law on well-being at work). The aim also is to promote health at work and a “well-being at work” culture. Often, the initial approach involves raising awareness and informing workers and direct supervisors about psychosocial risks and associated potential damage, like stress, burnout, and disengagement.

An important preventive trend to promote well-being at work, derived from positive psychology and the ‘Job Demands–Resources Model’ [59], aims to provide workers with adequate resources. This promotes optimal work engagement through personal fulfillment and development. Job resources can operate at different levels.

First, strategies usually implemented by human resource departments, especially practices related to recognition and feedback about job performance, development opportunities, and development of a work climate promoting trust and worker empowerment [69], can foster optimal worker fulfillment. For instance, Bakker [69] highlights that organizations implementing strategic and proactive approaches to work engagement can increase employee engagement.

Similarly, the role of the direct supervisor is crucial in promoting a positive team climate. Well-being at work will be higher if direct supervisors have the opportunity to lead discussion forums around job activities, and thus facilitate the regulatory work necessary for the team to function properly [70]. Several educational devices are aimed at developing skills, behaviors, and management practices conducive to mental health, such as tools/methods to help direct supervisors prevent psychosocial risks, training focused on work analysis, or training on broad management practices and managerial posture [71]. Additionally, communities of practice or ‘managerial practice exchange workshops’ are beneficial as they integrate the principles of psychosocial risk prevention and activity regulation into daily management practices [72,73].

Furthermore, the concept of ‘job crafting’ has recently emerged in work psychology [59]. ‘Job crafting’ refers to proactive changes made by workers themselves to modify their tasks, the type of relationships they maintain at work, or the meaning of their work [74]. It can also be defined as proactive changes made by workers regarding the demands and resources of their work [75]. Through their behaviors, workers can actively influence the provision of their job resources (e.g., asking for more feedback and help) and demands that represent personal challenges (e.g., engaging in a new project, mastering new skills), or reduction in demands that hinder fulfillment (e.g., reducing workload or bureaucratic aspects of work). These proactive changes allow workers to optimize their work environment by themselves. They also promote work engagement [76,77], job satisfaction and reduce burnout [77]. Tims et al. [77] demonstrated that job crafting interventions led to increased well-being, work engagement and job satisfaction and to a reduction in burnout among employees during the course of the study (two months). However, to stimulate these proactive behaviors, an organizational climate conducive to job crafting is essential. Some authors propose a multi-step organizational intervention to promote ‘job crafting’ behaviors among workers: (a) assess the strengths, motivations, and contributions of each team member; (b) synthesize each person’s tasks and duties; (c) consider distributing tasks and duties based on strengths and motivations; (d) assess possible changes in the work situation; (e) assess the consequences of the intervention; and finally (f) assess the obstacles and benefits of ‘job crafting’ [78].

Idealized work engagement can also sometimes lead to excessive work investment in a very fast work pace [79]. Work overcommitment is defined by a set of behaviors and attitudes that reflect the worker’s search for excessive recognition [80]. Based on a cross-sectional study on 634 participants, Preckel et al. [80] highlighted that overcommitted workers will, as a consequence, exaggerate their efforts and expose themselves to more job demands. This overcommitment was associated with an increase in vital exhaustion and a higher risk of burnout, underscoring the need for early intervention strategies [80]. Note that at this stage it is rare for the worker to confide in his/her general practitioner, prevention advisor, or psychologist. Indeed, the overcommitted worker is generally motivated and full of energy. This energy is also appreciated by the organization. Neither the worker nor their organization has a reason to complain about a job performance problem. A preventive approach would be to remain attentive to signs of worker hyperactivity and overcommitment at work. This attention can only be carried out by someone in direct contact with the worker (e.g., a team member or direct supervisor). This particular attention could, for example, be achieved through the development of a vigilance network, i.e., a device allowing the company to gain autonomy in considering difficult situations by professionalizing a reference team [81]. Therefore, addressing the signs of overcommitment in the training of network members, but also of direct supervisors or staff with a role in preventing psychosocial risks, would make it easier to identify workers at risk.

Finally, in terms of resources, one can also think of ergonomic solutions to promote adapted, comfortable, and secure work environments, or participatory projects around organizational values, which can foster organizational, but also team, identity [82].

Beyond job resources, primary preventive strategies can also apply to job demands. One of the principles of dynamic risk management is to eliminate or reduce psychosocial risk, so it is also necessary to reduce or limit job demands (e.g., inappropriate workload, task distribution-related problems, role ambiguity).

From the above, we can conclude that there are numerous and diversified primary preventions available for improving and promoting positive working conditions. As stipulated in the Belgian legislation on well-being at work (2014), each situation or work environment should be optimized through a prevention approach, including a diagnosis of psychosocial risks, an action plan directly resulting from the findings of the diagnosis, the implementation of the action plan, and its evaluation.

Table 1 summarizes the preventive actions to be prioritized according to primary prevention.

### 3.3. Key Moment 3: Weakening of the Ideal (Stage 1) and Protective Withdrawal (Stage 2)

We chose not to distinguish between stage 1 (weakening of the ideal) and stage 2 (protective withdrawal). Indeed, interventions relevant during the weakening of the ideal remain so during protective withdrawal. Individual-focused interventions are more likely to positively influence the worker early in the process, leading to burnout [83], i.e., from the first signs of exhaustion (Stages 1 and 2). However, these would be less effective for workers with confirmed burnout (Stage 3) [37].

Supporting workers at Stages 1 and 2 of burnout, as soon as the first clinical signs appear, can be carried out at the individual or organizational level [47], with the aim of job retention.

On an individual level, occupational clinic consultations allow for questioning of the ideal or of the causes leading to gradual loss, addressing the mismatch between expectations and work reality, especially through the psychodynamic work approach [84], or blocking the resources loss spiral and stimulating a gain in worker’s resources (Conservation of Resources Theory [85]). The Superior Health Council report [86] (p. 26) details the content of occupational health sessions in the treatment program for workers in the early stages of burnout (secondary prevention) proposed in Belgium by FEDRIS [60]:To listen and to discuss work experience, allowing for subjectivity and taking time to recognize an individual’s suffering at work;To help the individual to identify individual, collective (especially organizational), medical, and legal resources to regain control of his/her situation;To reflect on ways to restore work meaning in one’s life and adjust expectations to match work reality;To provide individual support and career guidance, helping the individual assess his/her career and identify future paths, or develop a new professional project;To determine the type of psychological support or guidance needed: either individual techniques, like cognitive, behavioral, emotional, or physical approaches (e.g., relaxation), or collective techniques, like ‘self-management skills’ training (e.g., stress management, energy recovery), or peer support groups;To discuss the opportunity for a multidisciplinary meeting within the workplace and, if necessary, prepare the worker for this exchange about working conditions in order to consider job retention (e.g., role clarification, responsibilities, type of direct supervisor support, workload, working hours) or return to work after sick leave (e.g., gradual return, working conditions’ adjustment, preparation of colleagues and the direct supervisor).

In addition to these sessions where work subjectivity is addressed, psychoeducation sessions to stimulate the worker’s self-care are also important early in the treatment program [87]. These sessions should focus on areas like healthy diet, energy recovery, sleep management, physical activity, and stress management.

As previously mentioned in the FEDRIS Burnout Treatment Program [60], individual-focused interventions can also include more specific components. These can focus on developing self-management skills like time management, assertiveness, conflict resolution, or negotiation, in order to enable the worker to manage stress situations [88]. The goal is to change or enhance resources and adaptive responses. Coping strategies can also be improved using cognitive and behavioral techniques. One of the objectives of these techniques is cognitive restructuring, especially in the case of work overcommitment, to identify irrational work-related thoughts and replace them with more realistic ones [80,89]. This restructuring can aim to reduce individual expectations, reinterpret the meaning of individual behaviors, clarify individual values, or consider new goals or a new perception of work role [90].

Furthermore, if overcommitment is a way for the worker to forget about mental doubts and anxieties, attention-diverting activities, like sports [47], meditation and mindfulness [48], relaxation [49], or even taking vacations [91], could reduce burnout symptoms. However, the benefit of these activities would be limited as, once they are over, the worker would return to painful and doubtful thoughts. It seems that, if the burnout score decreases during vacations, it returns to its pre-vacation level four weeks after returning to work [91]. Some attention-diverting activities might therefore be beneficial for the worker as long as they are performed daily. The term “psychological detachment” has been used to describe the process by which a worker distances himself/herself from his/her job outside of working hours [92], in the evenings or during weekends. Detachment outside of work has been associated with positive health outcomes [92]. Indeed, detachment stops the energy-depletion process and provides new resources for the worker. Without such detachment, the necessary recovery of the worker’s well-being would not occur when work leads to negative affects [92]. However, the problem remains; the worker’s ideal is still challenged. Doubt will persist as long as the ideal confronts contradictions in the work environment.

Hence, there is a need to combine these individual-focused interventions with organization-focused interventions. Although studies on organization-focused interventions are less common, workplace interventions are recognized as more effective [75]. Focusing on the work environment allows for addressing the root causes of the working condition problems and targeting a larger group for intervention. In a comprehensive study of 25 intervention programs implemented between 1995 and 2007 concerning burnout, some authors [54] distinguished interventions targeting the individual (17 out of 25 interventions), those targeting the organization (2 out of 25), and those combining both perspectives (6 interventions). Interventions focused on the individual often involve cognitive–behavioral interventions to increase competence, adopt “coping” behavior, and increase social support, or relaxation exercises. Organizational interventions often target how work is organized, leadership styles, changes in work procedures, and reduced task demands. Their study found that 80% of these interventions effectively reduced burnout in the short term (six months or less). These effects lasted longer (twelve months or more) if the intervention included measures targeting both the individual and the organization, but in all cases the positive effects of the intervention decreased over time. Continuous attention to burnout is therefore necessary to maintain positive effects [93].

Overall, the organization-centered approach allows for reflection on work-related demands, lack of job resources, or contradictions observed in work execution. This reflection should stimulate the search for solutions related to adapting to employment conditions or adjusting working conditions. Here, a multidisciplinary approach involving health professionals and prevention actors is often recommended. A participatory approach allowing group reflection (in a team, among professionals, etc.) on solutions to work-related problems seems effective in addressing job demands [94].

The transition from the weakening of the ideal (Stage 1) to protective withdrawal (Stage 2) typically highlights a decrease in perceived resources. The importance of resources is highlighted in two burnout models: the conservation of resources theory [85] and the job demands–resources model [59]. The former posits that, the fewer resources a worker has, the more vulnerable he/she is to lose his/her existing resources and the less able he/she is to acquire new ones [85]. The latter emphasizes the importance of resources to promote a motivational process, increase engagement, and mitigate the negative impact of job demands [59]. It, therefore, seems crucial to provide new resources or at least preserve those still present. Especially, promoting support from direct supervisors and colleagues, creating collaborative areas [47], or seeking more general social support [95] can intervene at this stage to increase workers’ resources. Mentorship can also be beneficial for newcomers or workers facing changes, in order to improve communication and support within the organization [96]. Finally, collective strategies can be interesting. One example can be the opportunity to participate in peer support groups (including with colleagues or direct supervisors) such as health professionals’ “Balint groups” [97]. Finally, organizations can propose employee assistance programs (EAPs), which are a voluntary workplace service providing free and confidential assessments, counselling, and support for personal or job-related concerns [98]. Their goal is to analyze difficult work situations in a group, express and share problems, break out of isolation, and let go of feelings of guilt [99,100,101]. This type of intervention is at an organizational level (as these support groups are organized within companies), but also at an individual level (as the intervention directed towards individuals’ feelings, rather than work organization).

Table 2 summarizes the preventive actions to be prioritized according to secondary prevention.

### 3.4. Key Moment 4: Burnout

The process of addressing confirmed burnout requires a multidisciplinary approach that includes health professionals (general practitioners, psychiatrists, psychologists), prevention actors (psychosocial prevention advisors, occupational physicians), and professional environment actors (HR managers, direct supervisors, union representatives, return-to-work coordinators).

The general practitioner often plays a frontline role in burnout management, especially in handling work incapacity (medical certificate), work resumption, and follow-up to prevent potential relapses. This long-term relationship involves active listening, which is crucial for decoding burnout symptoms. Sick leave should be considered based on the patient’s work and personal situation and his/her health condition. The general practitioner will aim to minimize or limit medication as much as possible (‘LESA/Landelijke Eerstelijns Samenwerkings Afspraak, National First-Line Collaboration Agreement in the Netherlands) [102]. However, during the initial period of sick leave, due to psychological complaints and severe sleep disorders, the general practitioner might prescribe benzodiazepines to address sleep issues and restlessness, as well as symptomatic medications for functional physical complaints, like pain and gastrointestinal issues. Mesters and colleagues recommend prescribing antidepressants only if the patient has a history of depression or suffers excessively from depressive symptoms [103].

Within the multidisciplinary approach, therapeutic treatment by an occupational clinician is essential. Several recommendations are available on this topic. The first, drawing from Evidence-Based Medicine Practice (based on international guidelines) and on the LESA guideline from the Netherlands, suggests a three-step follow-up [102,104]. The first step, the crisis phase, typically lasts for two to three weeks. Given that fatigue and exhaustion are central issues and functional capacity is hindered, it is essential to grant the patient rest. During this period, the goal is for the patient to rest and regain energy. It is also an opportune time to acknowledge the patient’s suffering, addressing work realities and associated emotions, allowing the individual to gradually accept his/her condition. The second step, the problem and solution analysis phase, spans the three to six weeks following the crisis phase. This phase requires active support to structure problems and seek solutions, in collaboration with the patient and his/her close relatives. During this stage, the patient will enhance and develop his/her adaptive skills, work on stress-inducing thoughts, strengthen positive emotions, and increase body awareness, self-confidence, and social relations. The final step, the solution application phase, also lasts between three and six weeks, aiming to guide and encourage the individual to implement solutions in his/her daily life, both professionally and personally. Typically, the person can return to his/her work environment within three months. If the patient still feels unable to work after four to six weeks, more support and other specialists are needed.

Health professionals will assess with the patient the need for additional psychological approaches, such as those related to Stages 1 (weakening of the ideal) and 2 (onset of protective withdrawal).

The occupational clinician can also gradually prepare the patient for a return to work by creating a psychological state conducive to work resumption, for instance, by boosting motivation, self-efficacy, and feelings of control, allowing them to set boundaries, or favoring a part-time return over no return [105]. Just as in actions for Stages 1 and 2, the patient should be prepared to negotiate adjustments to his/her working conditions or employment terms. For instance, Bataille [106] (pp. 67–68) suggests, based on qualitative interviews with prevention actors and burnout victims, key questions to support the worker in their post-burnout reconstruction. Questions about the centrality of work and professional identity are crucial in the perspective she proposes.

At this burnout stage, it might be necessary to support the individual in a career transition. This involves helping him/her identify resources to mobilize during the transition, the need for training, and consequently assisting him/her in accepting the time required for the transition. It also involves helping the person identify and cope with the social and/or financial consequences related to his/her status change.

Return-to-work support is a major issue in tertiary prevention. The multidisciplinary approach is essential: both prevention actors and work environment actors play a significant role in supporting the worker. Literature emphasizes three particularly important elements to consider for a successful return to work after a long sick leave: (1) a gradual work resumption, (2) improved or adapted working conditions, and (3) support from colleagues and direct supervisors [107].

According to Durand and Loisel’s therapeutic return-to-work model, the return should be progressive, starting with lighter tasks and increasing based on the worker’s capabilities [108]. Hochstrasser and colleagues also specify that it is preferable to increase the workload very gradually [109]. According to these authors, one way to determine an acceptable workload is to see how long the worker can perform an activity without interruption and without being excessively exhausted afterward. The workload can only be increased when the worker can sustain a certain energy level over time. Moreover, another study, conducted among workers with sick leave due to physical or mental health issues, shows that being able to return to part-time work is associated with a faster full-time return [110]. The authors emphasize that part-time return allows the individual to gradually expose themselves to work and serves as a positive experience [110]. Partially resuming work can help workers regenerate resources like improved self-confidence, a sense of self-efficacy and control, and thus promote a return to full-time work [86].

Regarding working conditions, returning to work after a long sick leave in the same work environment that contributed to the worker’s health depletion might cause a relapse [111]. Indeed, many workers fear the moment they will have to return to work because they are afraid of facing the same working conditions as before their work incapacity. Moreover, the worker might feel vulnerable regarding his/her ability to work efficiently. Thus, it is essential for the worker to see improvements/adaptations in his/her working conditions [107].

Lastly, social support is a significant motivator, especially if the worker feels respected by his/her colleagues and direct supervisor. Social support and the feeling of being welcomed by colleagues, with the specific contribution of the direct supervisor, are identified as crucial factors in the success of work resumption [110]. More specifically, the most important support for the worker seems to be related to expression of empathy and understanding [112]. The motivation for work resumption will be lower if the worker believes their colleagues and employer attribute the sick leave only to individual and not at all to work-related factors [113]. It is therefore important to sensitize the work environment about work-related factors impacting burnout, the core symptoms of the worker and the need for well-established return conditions [114]. Moreover, the direct supervisor plays a pivotal role in the quality of the return-to-work process. For instance, the return is more stable when the dialogue between the worker and his/her direct supervisor allows for determining the sick leave-related factors and the necessary adaptations for an efficient work resumption [50]. The willingness to cooperate in a return-to-work process varies from one direct supervisor to another. Direct supervisors, indeed, feel a conflict between their own objective of work performance and the adaptations requested by work resumption [105]. However, the direct supervisor’s attitude is influenced by the policy and attitude of the organization towards work resumption. Clearly defining the direct supervisor’s responsibilities in the return-to-work process increases his/her involvement [113].

Other key actors are also potentially involved in the work resumption process. Union representatives can intervene, although their role might be different from one organization to another. Their engagement may be limited by their role definition or by the hierarchy’s openness to their involvement in the process [105]. They rarely act on modifying the tasks to be performed. Instead, they ensure compliance with the return-to-work program. Good understanding between the hierarchy and union representatives facilitates work resumption [105]. Moreover, medical consultations with the occupational physician after a long sick leave can facilitate work resumption [50]. Finally, the human resources department plays a crucial role in managing long-term sick leaves. Its challenge is to reintegrate the worker under the best conditions. However, the return-to-work process is not always recognized as a human resources function [50], hence the interest in developing and implementing a policy for reintegrating workers after long-term sick leave and appointing a ‘return to work’ coordinator in the organization. It is therefore important for the worker to be aware of these key actors representing important resources to facilitate work resumption. A file explaining the steps and useful contacts during the return-to-work period can provide the worker with clear guidelines regarding the return-to-work process [50].

In their guide on return-to-work and job retention, St-Arnaud and Pelletier propose a seven-step approach to manage work resumption following mental health problems [115]. These steps seem to be adapted for work resumption after burnout. Since the approach proposed by these authors is described as an initiative or policy implemented by the company, several prerequisites are essential for the success of this approach [115] (pp. 9–10): (a) to debate the approach-related issues between professional actors (management, unions, and direct supervisors) in order to obtain their adherence to the approach; (b) engagement from the management committee; (c) sustained support from union representatives and (d) coordination of practices by internal and external actors. Once these conditions are met, the company should take various measures to implement this support approach. These measures will be adapted to the specific organization’s context [115] (pp. 12–17): (a) to create an implementation committee with different actors (employer, human resources services, union representatives, occupational physician, direct supervisor, specialists able to explain the clinical or social needs of the worker, interveners responsible for the approach); (b) to develop a framework of reference (common values and orientations of the approach); (c) to define the roles and responsibilities of each, including the coordinator of the approach; and (d) to develop and disseminate a communication and training plan. Following this, the return-to-work process can be implemented. The person responsible for the approach will have the role of accompanying workers from the beginning of their sick leave until their work resumption. This takes place in seven steps [115] (p. 19): (1) to initiate the approach from the first administrative procedures; (2) to establish the first contact with the worker (within 10 days of his/her sick leave); (3) to accompany the worker in his/her recovery; (4) to prepare the meeting with the direct supervisor; (5) to plan and develop a return-to-work plan; (6) to facilitate work resumption and implement the return plan; and (7) to ensure the follow-up of work resumption and make the necessary adjustments.

This structured return-to-work approach is supported by the study of Thomson and colleagues, which highlights several practices to facilitate work resumption after sick leave related to work stress [116]. According to these authors, it is important for the employer to maintain contact with the worker during his/her sick leave to develop a return-to-work plan with him/her. Furthermore, they emphasize the importance of modifying or removing potentially harmful work-related factors to prevent working conditions from getting worse after work resumption. Moreover, since psychosocial risks evolve over time, work resumption requires a long-term follow-up. Finally, the authors emphasize the importance of collaboration between professional actors and the worker in the return-to-work process. Indeed, this collaboration encourages the “empowerment” of the worker by allowing him/her to actively contribute to preparing his/her work resumption and regaining control over his/her professional life.

More recently, Karlson and colleagues evaluated the effect of a workplace-focused intervention on the sustainable return to work of people on long-term sick leave due to burnout [50]. Their results show that return to work is facilitated and is more common among people who benefited from the intervention compared to a control group. The intervention consisted of a convergence dialogue meeting. The aim of this meeting, held in the workplace, is to initiate dialogue between the worker absent due to burnout and the direct supervisor to find solutions to facilitate work resumption. During this meeting, the different points of view of the parties, their agreements and disagreements on the causes of sick leave, and the necessary changes to facilitate the return to work are discussed. The focus is mainly on the solutions and changes suggested, i.e., the search for converging perspectives and objectives between the direct supervisor and the worker. The idea is to identify ways to improve the fit between the abilities, expectations, and needs of the worker and the characteristics of the job. Lasting about an hour and a half, the meeting results in agreements regarding short and long-term solutions. Focusing on the fit between the job and the person is supposed to promote constructive communication between the worker and the direct supervisor. The authors advise implementing this worker-direct supervisor discussion approach at the beginning of the sick leave phase. Through their longitudinal study, Nieuwenhuijsen and colleagues also highlight the importance of good communication between the direct supervisor and workers absent due to mental health problems, in order to promote a full-time return to work [117]. However, their results demonstrate this positive impact of communication only among non-depressed or weakly depressed workers. The interaction between the severity of depressive symptoms and communication with the worker indeed suggests that the potentially favorable impact of communication is less important among workers with high depression scores. According to these authors, direct supervisors should communicate more frequently with absent workers due to illness and organize follow-up meetings more often to promote a faster work resumption.

Table 3 summarizes the preventive actions to be prioritized according to tertiary prevention.

## 4. Discussion

Previous research on burnout temporal stages establishes a clinical profile of a worker, detailing how positive work engagement can evolve into burnout [34,35,36]. This perspective facilitates the integration of prevention into practical applications. Indeed, the opinion review conducted on burnout prevention highlights the inherent complexity of this phenomenon, as well as the need to adopt a multi-faceted approach in definition and prevention. The complexity of the clinical picture and the specificity of each organizational context invite us to implement specific actions after an in-depth organizational analysis [118]. This perspective is useful to structure prevention strategies, adapted not only to the different stages of burnout but also to the wide variety of organizational contexts. This opinion review aimed to identify different types of interventions that could be relevant in the management of burnout, not in an exclusive manner, but inclusively, by integrating a combination of different approaches. The objective is, therefore, not to highlight the most effective intervention, but to propose a combined prevention strategy focused on both individual and organizational interventions, with the goal of establishing a tailored prevention plan according to the needs and possibilities of each situation.

The temporal dimension, central to our study, highlights the need to intervene at key moments in the burnout process [34,35,36].

According to this opinion review, we suggest that burnout primary prevention should not only concern workers but should already be integrated during early education (before the choice of studies and/or profession) and the personnel selection process. For example, sensitization sessions early in education are known to help raise awareness of psychosocial risks related to job’s realities [61]. This approach involves creating a network between education and field practice by engaging field professionals in the training process (e.g., through specific meetings or early observation internships in the training of future professionals) and by maintaining a teachers’ connection with the professional world. Similarly, in the hiring process, identifying the real expectations of job applicants and providing an overview of the organization as accurate as possible enables job applicants and the organization to make an informed choice about a future collaboration [62]. This perspective thus creates a well-balanced vision of the job and the working environment [61]. This approach allows professionals to identify a gap as realistically as possible between ideal and reality. In particular, this is known to increase job performance and to reduce attrition, some initial expectations, and turnover.

Secondly, direct supervisor involvement [71] is a key element for workers as far as primary prevention is concerned. Direct supervisors play a decisive role in creating a healthy work culture and applying effective prevention strategies. Their engagement in workers’ well-being, their ability to recognise early signs of burnout, and to respond proactively are essential to prevent burnout. This includes implementing flexible working policies, promoting work–life balance, and creating opportunities for workers to voice their concerns and actively participate in the design of their working environment. This last point refers to the concept of “job crafting”, highlighted by Bakker and Demerouti [59]. It calls for collective prevention by involving the organization as much as its workers. By taking the time to listen about the job challenges and demands (e.g., during team meetings) and by supporting workers to proactively adapt their task content and interpersonal relationships, direct supervisors can enhance their engagement and job satisfaction and also contribute to a reduction in the risk of burnout.

To further support practical application, it is essential to implement clear action plans that can be customized according to organizational needs. First, organizations should consider establishing structured feedback mechanisms that allow continuous dialogue between employees and management, ensuring that any signs of burnout are identified early. Moreover, offering regular training sessions for supervisors on how to manage and reduce work-related stress can be crucial. This could include modules on effective communication, conflict resolution, and promoting a supportive work environment. Organizations might also establish wellness programs that focus on mental health, providing resources, such as counselling services, stress management workshops, and relaxation spaces.

In addition, we observed that interventions implemented at an early stage, when workers begin to manifest contradictions and signs of stress while remaining engaged, can prevent the progression of burnout. These interventions are known as secondary prevention. This observation highlights the importance of organization-focused interventions to avoid a deeper deterioration in workers’ well-being, in compliance with the principles of the “Conservation of Resources Theory” [85] and Bakker and Demerouti’s “Job Demands–Resources Model” [59]. According to the latter model, a balance between job demands and available resources is necessary to prevent burnout. Indeed, secondary prevention can be seen as a proactive approach to rebalance these demands and resources as soon as the first signs of imbalance appear. For example, the organization can increase the number of resources available for workers, particularly through the support of colleagues and direct supervisors, and the creation of collaborative opportunities. At the same time, individual-focused interventions play an important role in cognitive restructuring [80,89,90] in addition to increasing personal resources, such as sport, meditation, and mindfulness, in order to promote psychological detachment [47,49]. Moreover, occupational clinic consultations, as described by Dejours and Gernet, aim to restore the match between expectations and the reality of work, by blocking the spiral of loss of resources and by stimulating their gain [84]. These approaches are both necessary and complementary, as they enable us to recognise and deal with the early symptoms of burnout, but also to treat the causes of suffering. These individual- and/or organization-focused interventions, by concentrating on individual needs and reinforcing available resources, can effectively prevent the progression to burnout. This confirms the importance of a combined approach based on primary and secondary prevention, focusing on both the organization and the individual to ensure an optimal and healthy working environment.

Finally, for organizations seeking to implement these strategies, it may be beneficial to create a task force or committee dedicated to mental health and burnout prevention. This group could be responsible for monitoring the effectiveness of interventions, providing regular updates to leadership, and ensuring that employees have access to the resources they need. Additionally, establishing metrics to evaluate the success of these interventions, such as tracking employee satisfaction, absenteeism rates, and productivity levels, could help in continuously refining and improving the burnout prevention strategy.

The treatment of burnout (Stage 3) highlights the need for a multidisciplinary approach, involving close collaboration between health professionals (general practitioner, psychologist, occupational clinician) and those involved in the workplace (occupational physician, psychosocial prevention advisor) at various times. This collaboration is important to ensure effective treatment of the burnout symptoms and to prepare the patient for a progressive return to work. For example, the role of the general practitioner is central, not only in the treatment of physical and psychological symptoms, but also in long-term follow-up, to prevent relapses [102,103]. The steps of therapeutic follow-up, founded on Evidence-Based Practice, based on international guidelines, and LESA, are fundamental to build a multidisciplinary care in distinct phases, from the crisis phase to the application of solutions [102,104]. In addition, it is essential to prepare the worker’s return to work, in line with the suggestions made by Baril and colleagues, especially by creating a favourable psychological environment and a progressive return to work [105]. Returning to work after burnout involves organisational adjustments to avoid relapse and the vulnerability associated with return [64,111]. For example, empathic support from colleagues and direct supervisors [112,113], communication between worker and direct supervisors [50], or the implementation of a well-defined and structured reintegration policy, as recommended by St-Arnaud and Pelletier [115].

### 4.1. Limitations

These recommendations for burnout prevention strategies are intended solely as a guideline and are not to be considered prescriptive or comprehensive in any way. The diversity of people suffering from work-related problems, in terms of profile, and the diversity of organizational models that have contributed to their suffering, prevents this problem from being dealt with by “standardized solutions”. Finally, we must also keep in mind that burnout is an evolving process, hence the importance of suggesting preventive measures, both individual and organizational, but also at primary (beforehand), secondary (when the first contradictions and symptoms appear) and tertiary levels (when the psychological or medical damage no longer allows professional activity). Our analysis also reveals significant gaps in current research, particularly concerning the sustainability and transferability of burnout prevention interventions. Demerouti and colleagues have highlighted this gap, emphasising the need for longitudinal studies to evaluate the long-term effectiveness of interventions and their applicability in various professional contexts [119].

### 4.2. Future Research and Practical Implications

Future research should examine how prevention strategies can be adapted and integrated into different organizational cultures and working environments. The question of prevention sustainability is especially relevant given that burnout is not a static phenomenon, as it evolves and changes according to working conditions and personal factors.

This paper proposes a comprehensive framework aimed at better understanding of how preventive actions can be synchronized with a temporal perspective of burnout to address this issue as early as possible. However, its purpose is not to evaluate the effectiveness of existing interventions, but rather to compile and structure them based on extensive research and field expertise. Therefore, future research could investigate, through meta-analysis or systematic review, articles presenting evidence of the effectiveness of various individual, organizational, and combined interventions concerning burnout.

Furthermore, our discussion highlights the need for a paradigm shift in the way burnout is approached in the workplace. It is essential to treat burnout not only as an individual problem, but also as an organisational and a societal issue. This broader perspective encourages us to reconsider our approaches to prevent burnout, by emphasising the creation of more supportive work environments and the importance of a healthy work-life balance.

## 5. Conclusions

Our opinion review contributes to the articulation of burnout prevention from a temporal perspective, structured around four pivotal stages in the burnout process. By including the three conventional forms of prevention (primary, secondary, and tertiary), this study review highlights the necessity for targeted interventions at each phase of burnout progression, from initial prevention that begins at the onset of a career, or even before, as one’s professional aspirations take shape (Stage 0), to interventions during the phase of confirmed burnout (Stage 3), including the early recognition of initial contradictions that causes the perception of work to deteriorate and undermines aspirations (Stage 1), and of the effects of these contradictions and coping strategies to reduce exposure to occupational stressors while conserving mental and emotional reserves (Stage 2).The complexity of burnout, with its multiple facets and implications, requires a holistic and personalised approach. This approach must be based on a deep understanding of both workers’ and organizational needs, as well as on an ongoing evaluation of the interventions’ effectiveness over time.

## Figures and Tables

**Table 1 ijerph-21-01617-t001:** Preventive actions according to primary prevention.

	Primary Prevention (Stage 0)
Key moment 1 Constructing the Ideal	- Raise awareness of the profession realities and risks during primary education - Provide a ‘Realistic Job Preview’ during personnel selection - Analyse the ‘Needs–Supply Fit’ during personnel selection
Key moment 2 Engagement and Enthusiasm with a High Job Ideal	- Develop a dynamic risk management policy - Promote health at work and a ‘well-being at work’ culture - Raise awareness/inform about burnout and psychosocial risks - Foster resources at work (e.g., (1) HR policies focused on recognition, feedback, development; (2) enhancing a positive climate; (3) ergonomic solutions for a pleasant working environment; (4) job crafting behaviour stimulation; (5) vigilance network implementation) - Prevent suffering at work by reducing or limiting work-related demands

**Table 2 ijerph-21-01617-t002:** Preventive actions according to secondary prevention.

	Secondary Prevention (Stages 1–2)
Key moment 3 Weakening of the Ideal and Protective Withdrawal	At an individual level, to offer, develop or promote: - Occupational clinic consultations - Psycho-education sessions for healthy lifestyle - ‘Self-management skills’ - A broader range of coping strategies - Activities that disengage attention - Individual consultations based on psycho-corporal and/or cognitive-behavioural approaches At an organisational level: - Reflect on work-related demands and search for solutions to reduce them - Provide new resources or at least retain existing ones - Employee Assistance Programmes (EAP) - Participation in peer support or discussion groups

**Table 3 ijerph-21-01617-t003:** Preventive actions according to tertiary prevention.

	Tertiary Prevention (Stage 3)
Key moment 4 Burnout	- Support patients on sick leave (General Practitioners—GPs) - Therapeutic follow-up by an occupational clinician (e.g., LESA and EBM follow-up in 3 stages, return to work or transition preparation) - Support the return to work (e.g., reintegration policy, workstation and working conditions adaptation, professional contacts’ implication such as colleagues, RH, manager)

## Data Availability

Not applicable.

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
