# Peer review of "Temporal Stages of Burnout: How to Design Prevention?"

_ijerph, 2024, doi:10.3390/ijerph21121617_

Round 1
Reviewer 1 Report (Previous Reviewer 1)
Comments and Suggestions for Authors
All reviewer comments have been thoroughly addressed. I look forwards to the publication of this manuscript.
Author Response
Thank you very much for your kind feedback and for your review throughout the process. We greatly appreciate your time and effort in helping us improve the quality of our manuscript. We are delighted to hear that you look forward to its publication.
Reviewer 2 Report (Previous Reviewer 2)
Comments and Suggestions for Authors
Very good job updating this submission and significantly fleshing out your submission. Apart from few language typos/editing and points of emphasis this is a great addition to the literature. See the attached tracked comments.
Notably:
- Clarify your study objective at the end of the introduction to focus the reader on the goal of the paper (Such as reiterated in the discussion.
- Please review the highlighted points in the abstract. I believe you introduced some errors in some sentences in your update.
- Do add additional check for typos without the track changes function.
- Reconcile the description of the report: lit review vs. opinion review, etc.

Author Response
Very good job updating this submission and significantly fleshing out your submission. Apart from a few language typos/editing and points of emphasis, this is a great addition to the literature. See the attached tracked comments.
Notably:
- Clarify your study objective at the end of the introduction to focus the reader on the goal of the paper (such as reiterated in the discussion).
- We clarified the study objective at the end of the introduction (Lines 253 to 259).
- Please review the highlighted points in the abstract. I believe you introduced some errors in some sentences in your update.
- Lines 16 to 19: We clarified the sentence:
“The review criteria allow for the integration of both individual- and organizational-focused interventions, ranging from early organizational-level strategies (primary prevention) to clinical consultations addressing the erosion of professional ideals (secondary prevention), as well as psychoeducational sessions aimed at promoting worker well-being.” - We removed the following passage:
“From primary prevention to tertiary prevention, results showed primary prevention from the outset and even upstream of one’s career (Stage 0); secondary prevention to confront initial contradictions and the weakening of the professional ideal (Stage 1) and during the protective withdrawal (Stage 2), and finally, tertiary prevention during confirmed burnout (Stage 3).” - We deleted the second “In conclusion” and kept it only at line 19.
- We revised the first sentence of the limitations section (Line 887):
“These recommendations for burnout prevention strategies are intended solely as a guideline and are not to be considered prescriptive or comprehensive in any way.” - Do add an additional check for typos without the track changes function.
- We reviewed the article and identified several typos and errors:
- Line 128: “aim at identify the burnout” → “aim at identifying the burnout”
- Line 218: “cfr.” → “cf.”
- Line 327: “job vacancy” → “job vacancies”
- Lines 440 & 566: “Tables” → “Table”
- Line 447: “the stage 1” → “stage 1”
- Line 527: “targete” → “target”
- Lines 559 & 830: “counselling” → “counselling”
- Line 647: “show” → “shows”
- We thank you for your review and hope to have identified all remaining typos.
- Reconcile the description of the report: lit review vs. opinion review, etc.
- We standardized the terminology, using “opinion review” consistently throughout the manuscript (Lines 14, 776, 924).
This manuscript is a resubmission of an earlier submission. The following is a list of the peer review reports and author responses from that submission.
Round 1
Reviewer 1 Report
Comments and Suggestions for Authors
Thank you to the authors for this work. The clear description of the temporal stages of burnout will be of value both to clinicians, managers and researchers, as it will enable a targeted intervention approach to be taken. The authors are clearly extremely familiar with the burnout literature, with their accurate reference to a variety of burnout models and concepts such as ‘job crafting’.
The authors make a good argument re using a narrative review. The holistic approach and wide-ranging perspectives are apt and practically useful, for instance, signaling what interventions might be useful for an HR manager or a clinical supervisor. The tables of prevention strategies are well-written and practically useful.
A few minor comments and suggestions are below:
Line 22: I am not sure that it needs to be narrowed to ‘employees’ facing burbout. This work will be valuable for many wishing to tackle burnout including those who are self-employed
Line 24: Would they wish to add concepts re ‘BO intervention’ or ‘BO management’ or ‘return-to-work’ or similar to the keywords?
Lines 26-37: In the introduction where the authors highlight the importance and significance of burnout, they may wish to also allude to the fact that burnout goes beyond an individial and seeps into the team – ie. the effects on the work unit. And it may also be worth furhter strengthening this section by highlighting the consequences of not addressing it - with perhaps a reference to risks (e.g. error/quality of service/productivity etc) alongside attrition and employee ill-health.
In Section 1.1 where the definition of burnout is discussed, along with the fact that it is not a diagnosis, the authors may also wish to note that Burnout is included in the International Classification of Diseases 11, (ICD-11, Jan 2022) where it is classified as an occupational phenomenon, not a medical condition.
In Section 1.1 the authors may also wish to consider mentioning that although there are these temporal stages of burnout, the person who ‘experiences’ the burnout may not be aware of the stages as they progress. For some people, burnout feels like a ‘sudden’ thing that has happened – because they may have a sudden realisation of their mental and physical state (exhaustion/’can’t go on’ etc), whereas in fact, it has been building up over many months.
Lines 125-137: the authors rightly point out that using individual and organisational approahces together is the most effective way. As well as this, in this section, they may wish to highlight the difference in effectiveness of using solely organisational appraoches or solely individual appoaches, as well as making some comment on how sustained any changes might be from interventions that have been tried in the literature. I note that the authors do include an excellent discussion of this towards the end of manuscript (lines 388-404), however it might be worth adding one sentence re this in the introduction.
Line 291: It is pleasing to see the authors noting issues such as help-seeking behaviour, which have an impact on the uptake of certain interventions, something which is not always acknowledged.
Reviewer 2 Report
Comments and Suggestions for Authors
This is a very well written paper. I particularly appreciate the work put in to formalize such an amorphous area of public and worker health.
A few potential errors noted on lines 191 ("legislations"); and 313/314 (check referencing).
It would also be helpful to report the process for selecting/excluding reviewed articles (preferably in form of a flow chart), as well as include the articles in a summary table to delineate from additional referenced articles.
Reviewer 3 Report
Comments and Suggestions for Authors
The Manuscript „Temporal Stages of Burnout: How to design prevention?“ explores prevention strategies which can be applied at different stages of burn-out. In general, the idea and the topic of the Manuscript is very good and important, and the Manuscript is very well written, underlying an important problem and potential solutions for it. I believe this would be very useful for the readers of IJERPH.
Nevertheless, the Manuscript lacks structure of a scientific article in its most important parts (methods, results) and should be improved in that direction before it can be considered for publication. Since the Authors are using the format of a narrative review, it would be important to give direction to future research which could focus more on evidence regarding burnout prevention strategies.
Abstract
The abstract is too general, and there is no distinction between the introductory statements, no methods are mentioned (how were the papers searched fore, retrieved, screened, etc.) which is necessary for a review article. No actual results are presented, and no actual conclusions which could be drawn from the results.
I think the Authors need to explain the structure of the Manuscript at the beginning, right after the Introduction.
There is no clear Aim statement or research question asked.
Introduction
First paragraphs: very general, especially Lines 27-32. Who says and what data shows that there are market fluctuations, turbulent times, that wellbeing of workers is important?
The section 1.1.Definitions and temporal stages of burnout is crucial for this review, and the Authors base the stages of burnout on the same set of references (18-20) which should be described in more detail – how was this model developed, was it validated, etc… Then the description of the stages of burnout can follow.
Line 170: what is a dual approach? Do you mean working at individual and organizational level? It should be clarified
Methods
Methods cannot be part of the Introduction (currently 1.3 Methodology). This should be a separate section explaining all the methods used in this research to facilitate reproducibility and repeatability.
The Authors say they use a narrative review approach and cite a reference which is unfortunately not in English, so it is impossible (for me) to check the approach.
I would suggest using something more “international” such as the Baethge, C., Goldbeck-Wood, S. & Mertens, S. SANRA—a scale for the quality assessment of narrative review articles. Res Integr Peer Rev 4, 5 (2019). https://doi.org/10.1186/s41073-019-0064-8
Lines 177-182: The description of the methods here is too relaxed for a review. I understand the idea was not to write a systematic review, but maybe this then needs to be explained and justified well. Current description of the methodology prevents any reproducibility. Also, using only Google Scholar for the search seams kind of limiting, as it depents a lot on their algorithms and there is no guarantee you have included all the relevant articles.
Lines 183-191: Also the screening and selection process is very relaxed, including the specific inclusion and exclusion criteria.
Section 2 – Key moments
In general, this section is very well written. My comments are generally connected to the information provided about various interventions and the way they were provided.
It would bring much more value for the readers if the Authors could provide some kind of quantitative explanation of the results of the studies presented in the review. For example, lines 221-223 discuss “positive results in the UK” but there is no quantitative information about what was achieved (e.g. 2-fold reduction in risk from burnout during 5 years, or 50% lower prevalence of burnout in the intervention group compared to others). This should apply to all the proposed interventions. In case this is impossible, the Authors should try to find another way to give relevant information about the chosen interventions so that the reader could decide if this is worth doing or prioritize interventions by effectiveness and efficacy.
Tables are of low quality and should be improved (resolution). No need to have the tables as images.
Discussion
The Discussion is well written, but I believe that the Authors could provide their view, based on the literature review, regarding the priorities or effectiveness of various measures proposed and in which settings.
Limitations and future research
Since the Authors have decided to do a narrative review, they could inform future research by underlining what kind of systematic reviews (and meta-analyses?) could be done based on their research – to provide evidence regarding the various interventions.
Reviewer 4 Report
Comments and Suggestions for Authors
From the perspective of job burnout, the author elaborates on the characteristicsFrom the perspective of job burnout, the author elaborates on the characteristics of job burnout through the initial stage of work ideal participation and enthusiasm to the third stage of confirming burnout. This has important guiding significance for guiding employees' sense of participation in real production work. The article is innovative in its intention. Suggestions are as follows:
1. The content of the table in the article is vague, it is suggested to reorganize it;
2. It is suggested to add a flow chart to more intuitively express the author's viewpoint;
3. It is suggested to introduce methods and techniques for preventing and alleviating job burnout to help readers better cope with job burnout issues;
4. It is suggested to provide some practical suggestions and action plans at the end of the article to help readers apply what they have learned to their actual lives.